# Measurement of Charge and Refractive Indices in Optically Trapped and Ionized Living Cells

**Endris Muhammed** [1], **Daniel B. Erenso** [2], **Ying Gao** [3], **Li Chen** [3], **Michele Kelley** [2], **Carina Vazquez** [2], **Mitchell Gale** [2], **Cody Nichols** [4] and **Horace T. Crogman** [4,*]

1   Department of Physics, Addis Ababa University, Addis Ababa 1176, Ethiopia
2   Department of Physics, Middle Tennessee State University, Murfreesboro, TN 37132, USA
3   Department of Biology, Middle Tennessee State University, Murfreesboro, TN 37132, USA
4   Department of Physics, California State University Dominguez Hills, Carson, CA 90747, USA
*   Correspondence: hcrogman@csudh.edu

**Abstract:** The post-ionization dynamics of chemo-treated and untreated 4T1 breast cancer cells ionized by laser trapping techniques are studied. We have determined each cell's charge and refractive index by developing a theoretical model for the forces determining the post-ionization dynamics. The shift in a cell's refractive index due to an intense oscillating electric field was studied, and the results are reported here. We observed that a trapped cell, as it becomes charged, will eventually exit the trap perpendicular to the beam's direction; this means that the electric force of the cell overcomes the trapping force. As a result, the cell's conductivity changes due to the oscillating field, causing a decrease in the cell's refractive index.

**Keywords:** breast cancer cells; laser trapping; cell mechanics; chemotherapy; radiotherapy

## 1. Introduction

According to the International Agency for Research on Cancer (IARC) 2018 report, every year, approximately 2.1 million new cases and half a million breast cancer deaths occur worldwide [1,2]. The incidence of breast cancer is second only to lung cancer (12%), but its death rate is relatively low (7%). The higher survival rate might be credited to the advancement of traditional therapies such as radiotherapy [3,4] chemotherapy [5–8], surgery [9], and the development of newer therapies such as hypothermia and hyperthermia using nanotechnology [10–13] and immunotherapy [14].

Radiotherapy (RT) is one of the most effective tools used for different malignant tumors. It is used to kill cancer cells in a tumor with radiation energy powerful enough to overcome an atom or molecule's electron-binding energy. In RT, when such radiation energy is used to treat a patient, it typically causes damage to the normal cells surrounding the tumor. Therefore, the goal of RT must be to maximize the radiation damage to the cancer cells while minimizing the impact on normal cells. A combination of radiation and chemotherapy is known to be more effective than RT alone, because these treatments inhibit DNA repair processes, thereby enhancing the death of tumor cells [8]. Some chemotherapy drugs may increase the number of tumor cell clonogens that are susceptible to ionization energy, reducing the radiation dose needed to kill the tumor cells. To improve therapeutic outcomes in radiation and chemo combined modalities, platinum and taxanes have been used as standard chemotherapeutic agents. However, using these agents in that combination limits the radiation dose because these agents are considerably toxic to normal tissues [8].

Other recently developed combined treatment modalities involve the less-invasive hypothermia, hyperthermia, and biocompatible nanoparticle methods. For example, a recent study of 4T1 breast cancer cells in mice showed that hypothermia and hypoperfusion effects induced by paclitaxel (PTX) and maintained by reducing body temperature can

prevent tumor relapse or metastasis after chemotherapy [11]. The 4T1 cell line is an attractive model for human breast cancer due to its ability to be transplanted into mammary glands, its rapid spread to surrounding lymph nodes, and its persistence following the removal of the primary 4T1 tumor [15]. Additionally, biocompatible nanoparticles such as gold silica nanoshells (GSNs) have been used to treat prostate cancer [13] since they can absorb light in the infrared range, which has high tissue transparency. That ability in GSNs allows them to generate heat, which induces highly localized hyperthermia, a highly effective modality for photothermal cancer therapy. Thus, strategies for combined modalities of cancer treatment that utilize radiotherapy, chemo, and possibly hyperthermia effects could provide a new approach for better treatment efficacy.

Some studies have shown that oligostilbenes (naturally occurring compounds) used to treat various types of cancer contain antitumor agents that could increase the radiation sensitivity of tumor cells and provide protection against radiation-induced damage in normal tissues [16,17]. This study used a high-power infrared laser (at a wavelength of 1064 nm) to assess the radiosensitivity of a 4T1 breast carcinoma cell line treated with oligostilbene: 2-Dodecyl-6-methoxycyclohexa-2, 5-diene-1, 4-dione (DMDD) isolated from the root of Averrhoa carambola L [18–20]. DMDD treatment facilitates membrane breakdown, as measured by the threshold ionization energy [20]. The membrane breakdown builds up more charges on the cell; thus, the electrostatic force becomes increasingly stronger until it overcomes the gradient trapping and drag forces [20–24]. The charge resulting from such membrane breakdown in the untreated vs. treated is unknown.

The fact that DMDD treatment damages the cell membrane means that other biophysical parameters are impacted. For example, the refractive index is an important biophysical parameter that has been extensively studied. As detailed below, it can be determined and correlated with other biophysical cells' parameters, such as dry mass, wet mass, protein concentration, elasticity, conductivity, etc. Aside from studying cell division, infection, and radiosensitivity, this index can also be used to study certain metabolic processes [25]. This means that, in addition to representing intracellular mass and concentration, it also provides insight into various biological models and is correlated with other biophysical parameters such as mechanical, electrical, and optical properties [26].

A cell, under highly intense radiation, becomes charged. The post-ionization dynamic quantities such as displacement, velocity, and acceleration depend on the charge. Therefore, determining the changes in the cell's refractive index could be very important data to gather and advance treatment efficacy. Very little is reported about the behavior of cells' refractive indices under a high-intensity electric field. Therefore, further comprehension of biomedical optics must be determined by investigating such changes in cancer cells.

This article explores the relationship between a cell's charge and its refractive index. We wished to determine the magnitude of the charge and refractive index developed in the untreated, 2 h-treated, and 24 h-treated cell clusters by analyzing the post-ionization dynamics of each individual cell. To calculate the charge developed in each cell, we used two different approaches. The first approach assumes an unknown electrical susceptibility for both the treated and untreated cells, and the second approach assumes and uses the same electrical susceptibility for the treated and untreated cancer cells. Below, we discuss, compare, and contrast the methods and results obtained using these two approaches.

## 2. Methods

The experimental methods involve cell culturing, treatment by DMDD, and trapping by a high-power infrared laser. Untreated 4T1 breast cancer cells were compared to two groups treated with DMDD for two or twenty-four hours (as a control group). In this section, we discuss the methods involving the theoretical model we developed to describe the post-ionization dynamics to determine the cells' charge and refractive indices.

## *2.1. Cell Culture and Treatment*

4T1 cells were cultured in RPMI1640 medium with 10% FBS in a 5% $CO_2$ and 37 °C incubator. Cells were trypsinized and passaged every 2–3 days. After 4T1 cells were trypsinized, they were diluted with RPMI1640 medium, and seeded in a 96-well plate with an intensity of 5000–7000 cells per well (100 μL/well). After the cells were attached to the bottom of the wells for 24 h, they were treated with DMDD at 100 μM for 2 or 24 h. Each of the untreated group, 2 h treatment group, and 24 h treatment group had six replicate wells. As soon as the treatment was completed, the culture medium in each well was transferred into an Eppendorf tube. Following PBS rinsing, 50 μL trypsin was added to each well, and the detached cells were transferred to the same Eppendorf tube. These methods are also detailed in ref [20–24].

## *2.2. Laser Trap Set-Up*

The setup for the laser trap is shown in Figure 1. This experimental setup is very similar to the setup used in previous biomedical laser trapping application studies [20–24]. The laser has a wavelength of 1064 nm and a maximum power of 8 W. The power is controlled by a half-wave plate (W) and a polarizer (P). A 20X beam expander and a pair of 5 cm and 20 cm focal length lenses (L1 and L2) take the beam directed by mirrors M1 and M2 and expand it to about the diameter of the window of the microscope's objective lens (~2 cm); this expansion is critical for a stronger trap. At the microscope's focal plane, the mirror M5 created a steerable trap by directing and aligning the beam with the mirrors M3 and M4. M5 was placed 20 cm away from the third converging lens (L3), which is positioned 40 cm from another converging lens (L4) with the same focal length of 20 cm. L4 is placed 20 cm from the back of the objective lens. For a steerable trap to form on the focal plane of the microscope, L3 and L4 must be separated by twice their focal length. A Dichroic mirror (DM) positioned at 45 degrees inside the microscope coupled the collimated and aligned beam to the microscope. Assuming a normal incidence, the DM reflects the laser beam through a 100X objective with a 1.25 numerical aperture. A PC-controlled digital camera integrated into the microscope receives the imaging light from the DM via the second port of the microscope. At the same time, the DM transmits the imaging light from an Olympus T4 halogen lamp. For experiments conducted in trapping and ionizing a cell, the power was measured at two positions. One position is before the beam is transmitted to L4 (~4.34 W), and the other is after exiting the objective lens (~0.806 W). An efficiency of about ~18.57% was maintained throughout our measurements.

A well-slide containing 4T1 cells from the untreated control, 2 h-, and 24 h-treated groups was mounted onto a microscope micro-driven mechanical stage. As the cells were lying on the bottom of the slide, we used the digital camera to capture an image. Our next step was to open the laser port of the microscope and trap the cell inside. The digital camera took successive image captures of the ionized cell at a fixed frame-grabbing rate until the cell had been ejected from the trap and disappeared.

## *2.3. The Forces*

Cells suspended in FBS are trapped by the laser gradient force until they become ionized, and an electrostatic force is generated that forces them out of the trap. The instant the cell gets ejected, it also experiences a drag force. Thus, the post-ionization dynamics are determined by Newton's equation of motion,

$$m\frac{d^2\vec{r}}{dt^2} = \vec{F}_e - \vec{F}_d - \vec{F}_t \tag{1}$$

where $m$ is the mass, $\vec{r}$ is the post-ionization position of the charged cell from the center of the trap, and $\vec{F}_e$, $\vec{F}_d$, and $\vec{F}_t$ are the electrostatic, the drag, and the trapping forces, respectively.

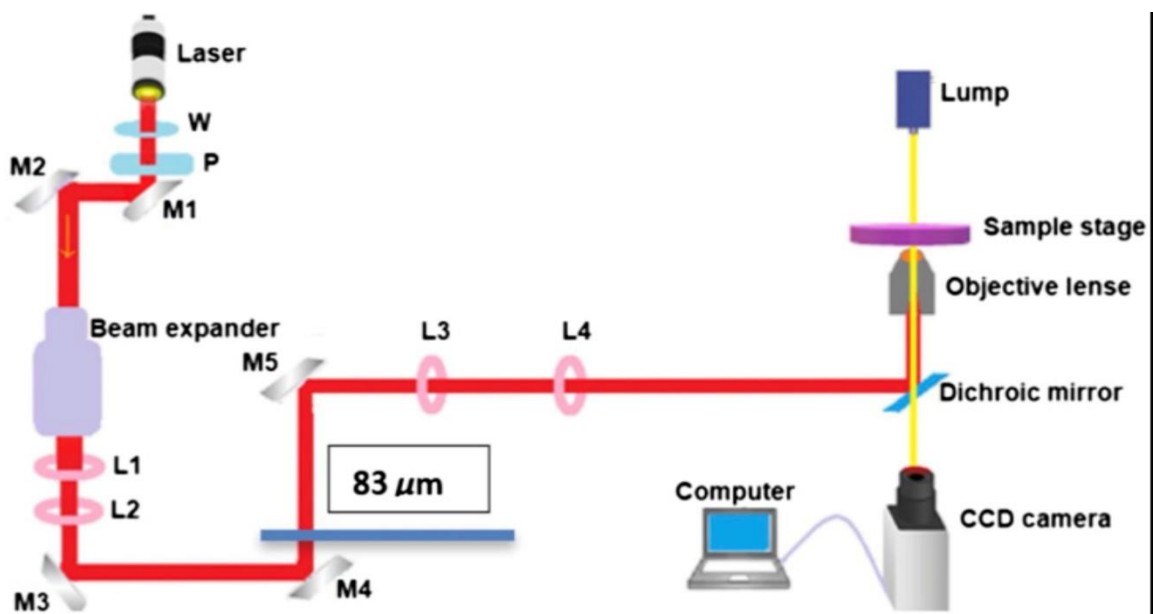

**Figure 1.** The experimental setup for the laser trap is shown in the diagram. It has a wavelength of 1064 nm and can output up to 8 W. Half-wave plates (W) and polarizers (P) control the output power. See [21]. for further detail.

The electrostatic force: As the cell is being ionized by the electromagnetic fields, there is a buildup of charge caused by the breakdown of the membrane. The free charge density depends on both position and time and can be represented by $\rho\left(\vec{r}, t\right)$. Let the magnitude of the electric field for a laser beam polarized along the $\hat{s}_0$ direction on the x–y plane and propagating in the z-direction, as shown in Figure 2a, be $E\left(\vec{r}, t\right)$. Then, the corresponding magnetic field for this laser beam is given by

$$\vec{B}\left(\vec{r}, t\right) = \frac{n}{c} E\left(\vec{r}, t\right)(\hat{z} \times \hat{s}_o) \tag{2}$$

where $n$ is the refractive index of the cell and $c$ is the speed of light in a vacuum.

It is important to note that the refractive index of the cell, $n$ in Equation (2), depends on time, as the cell is undergoing dielectric breakdown due to the ionization taking place while the cell is in the trap. Suppose a free charge, $dq'$, has developed in an infinitesimal volume, $dV'$, of the cell, which we can express in terms of the free charge density as

$$dq' = \rho\left(\vec{r} + \vec{r}', t\right) dV' \tag{3}$$

and this charge has a velocity,

$$\vec{v}' = \frac{d\left(\vec{r} + \vec{r}'\right)}{dt} = \frac{d\vec{r}'}{dt} = v'\hat{r}\prime \tag{4}$$

Note that we have neglected the change in the position of the center of mass of the cell $\vec{r}$ while the cell is in the trap; thus, the Lorentz force on the total free charge of the cell can be determined using

$$\vec{F}_e\left(\vec{r}, t\right) = \int_V \rho\left(\vec{r} + \vec{r}', t\right) E\left(\vec{r} + \vec{r}', t\right) \left\{ \hat{s}_o + \frac{nv'}{c}\left(\hat{r}' \times \hat{z} \times \hat{s}_o\right) \right\} dV' \tag{5}$$

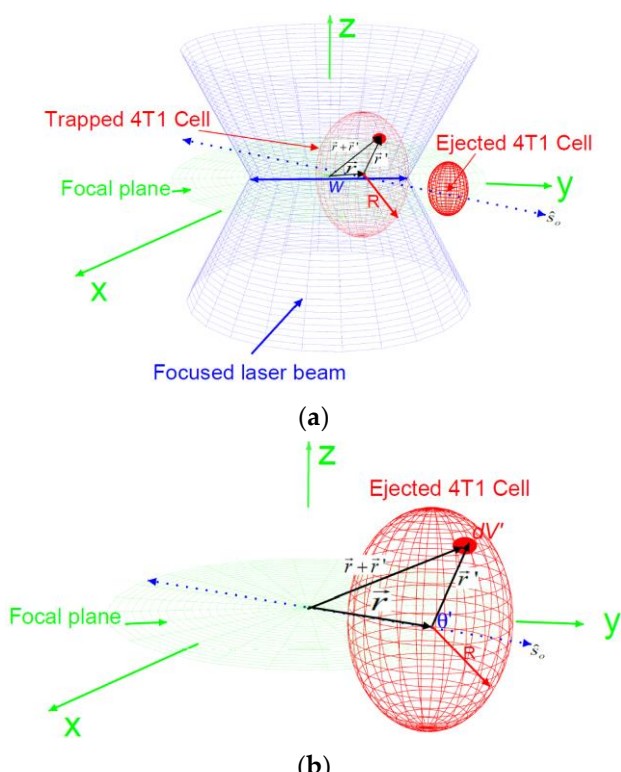

**Figure 2.** (**a**) The blue double cone is a schematic for a focused linearly polarized laser beam along the direction $\hat{s}_o$ on the x–y plane and propagating in the z-direction. This beam has a diameter of *2w* at the trap location. The red sphere represents a 4T1 cell before ejection (big) and after ejection (small). The vector $\vec{r}$ is the position of the center of the cell as measured from the center of the trap; $\vec{r}'$ is the position of an infinitesimal free charge, $dq'$, from the center of the cell. (**b**) A schematic for the position at a given instant in time for an ionized and ejected 4T1 cell along the direction of polarization $\hat{s}_0$; $\theta$ is the angle between the polarization direction and the vector $\vec{r}'$.

Next, we make two physically valid approximations. In the first approximation, we neglect the magnetic contribution to the Lorentz force because the speed of the free charge, $V'$, is negligible compared with the speed of light, $c$. The second approximation that we make is for the time-dependent charge density $\rho\left(\vec{r},t\right)$; we are supposing that the time-lapse from the moment the cell gets trapped to the moment it got ejected is $T$. During this time, a significant amount of the free charge is created when the membrane is significantly ruptured, which happens at time $t = t_0$. If we neglect the free charge developed prior to and after this time and represent the total charge on the cell by $q_0$ at the position of its center of mass, $\vec{r}$, we may approximate the charge density using the Dirac delta function,

$$\rho\left(\vec{r} + \vec{r}',t\right) = q_o\delta\left(\vec{r}' - \vec{r}\right)T\delta(t - t_0) \tag{6}$$

Then, the time average Lorentz,

$$\vec{F}_e\left(\vec{r}\right) = \frac{1}{T}\int_0^T \vec{F}(t)dt \tag{7}$$

under these two approximations,

$$\vec{F}_e\left(\vec{r}\right) = \left(q_o \int_v \int_0^T E\left(\vec{r} + \vec{r}',t\right)\delta\left(\vec{r}' - \vec{r}\right)\delta(t - t_o)dV'dt\right)\hat{s}_0 \tag{8}$$

becomes

$$\vec{F}_e\left(\vec{r}\right) = q_o E_o\left(2\vec{r}, t_0\right)\hat{s}_o \tag{9}$$

For a Gaussian Electromagnetic wave propagating in the positive z-direction, the electric field can be expressed as

$$E\left(2\vec{r}, t_0\right) = E_o exp\left[-\frac{r2}{w^2}\right]exp[-i(kz - \omega t_0)] \tag{10}$$

where $w$ is the beam radius at the trap location. For the post-ionization dynamics, we set the initial time at $t = t_0 = 0$, such that the cell's center of mass at this initial time is at the origin, and we assume that the cell remains confined to the x–y plane throughout its post-ionization motion so that z = 0. Setting these values in Equation (10) and substituting the resulting equation into Equation (9), the electrostatic force is given by

$$\vec{F}_e\left(\vec{r}\right) = q_o E_o exp\left(-\frac{r^2}{w^2}\right)\hat{s}_o \tag{11}$$

The amplitude of the electric field, $E_0$, can be determined from the power, $P$, measured at the trap location using

$$E_0 = \sqrt{\frac{2Pv\mu_0}{A}} \tag{12}$$

where $\mu_0$ is the magnetic permeability of free space, $v$ is the speed of light in the cell's suspended medium, and $A = \pi w^2$ is the beam size at the trap location determined from the beam radius at the back of the objective lens and its numerical aperture [25].

The drag force: As we saw in Figure 2b, the 4T1 cancer cells are modeled spherically in shape with radius $R$, and the drag force can be determined using

$$\vec{F}_d = 6\pi\mu R\frac{d\vec{r}}{dt}, \tag{13}$$

where $\mu$ is the viscosity of the FBS fluid the cell is suspended in.

Trapping force: We use electromagnetic energy (EME) change in the region occupied by the cell to estimate the trapping force on the cell. To this end, let us consider a cell (a dielectric sphere) with radius $R$. After the cell is ejected, as shown in Figure 2b, let the position of the center of mass of the cell from the center of the trap, assuming the cell is confined on the x-y plane, be $\vec{r}$, which is directed in the direction of the polarization of the field $\hat{s}_0$, We consider an infinitesimal volume, $dV'$, inside this cell at a position $\vec{r}'$ that makes an angle $\theta'$ as measured relative to the direction of the vector $\hat{s}_0$, as shown in Figure 2b. The electric field of a Gaussian laser beam propagating along the z-direction at the position of the infinitesimal volume can then be expressed as

$$\vec{E}\left(r, r', \theta\right) = E_0 exp\left[-\frac{1}{4w^2}\left(r^2 + 2rr'cos\theta' + r'^2\right)\right]\hat{s}_0 \tag{14}$$

where $E_0$ is the amplitude of the field, which is calculated from the power, and $w$ is the beam radius of the laser at the trap location.

As we stated earlier, we are interested in finding the trapping force using the electromagnetic energy change in the micro space occupied by the cell. Thus, one must find the energy before and after the cell is exposed to the laser field. In the micro-volume, $V$, with electrical permittivity constant $\varepsilon_b$ (space which later is occupied by the cell), the electric field and the corresponding electric displacement of the laser field are $\vec{E}(r, r', \theta,)$ and $\vec{D}(r, r', \theta,)$, and the electromagnetic energy in this volume of space, $W_b$, can be determined using

$$W_b = \frac{1}{2}\int \vec{E}\left(r, r', \theta'\right)\vec{D}_b\left(r, r', \theta'\right)dV' \tag{15}$$

Similarly, the energy after the cell, with electrical permittivity constant $\varepsilon_a$, occupying this same volume of space, is

$$W_a = \frac{1}{2} \int \vec{E}(r, r', \theta') \vec{D}_a(r, r', \theta') dV' \tag{16}$$

Then, the change in EME in the micro-volume, *V*, occupied by the cell is given by

$$\Delta W = W_a - W_b = (\varepsilon_a - \varepsilon_b) \frac{1}{2} \int E^2(r, r', \theta') dV' \tag{17}$$

where we used $\vec{D} = \varepsilon \vec{E}(r, r', \theta,)$ for a linear medium. Then, using the electric field in Equation (14), we write

$$\Delta W = \tfrac{1}{2}(\varepsilon_a - \varepsilon_b) E_o{}^2 \int_0^R \int_0^\pi \int_0^{2\pi} exp[-\tfrac{1}{4w^2}(r^2 + 2rr' \cos\theta' \\ + r'^2)] r'^2 dr' \sin(\theta') d\theta' d\varphi' \tag{18}$$

Upon integrating this equation, we find

$$\Delta W = (\varepsilon_a - \varepsilon_b) \pi E_o^2 \frac{w^3}{2} [-\frac{w}{r}(e^{-\frac{(r_t+R)^2}{2w^2}} - e^{-\frac{(r_t-R)^2}{2w^2}}) \\ + \sqrt{2\pi}(Erf[\frac{1}{w}(R+r)] + Erf[\frac{1}{w}(R-r)])] \tag{19}$$

The trapping force is given by

$$\vec{F} = -\nabla_r(\Delta W) \tag{20}$$

and using Equation (20) is found to be

$$\vec{F}_T(r) = -8(\varepsilon_a - \varepsilon_b)\pi \frac{E_o^2 w^4}{r^2} e^{-\frac{(r^2+R^2)}{2w^2}} \left[ \frac{Rr}{w^2}\cosh\left(\frac{Rr}{w^2}\right) - \sinh\left(\frac{Rr}{w^2}\right) \right] \hat{r} \tag{21}$$

or, in terms of the refractive indices of the cell $\varepsilon_b = \varepsilon_0 n_b^2$ and the medium $\varepsilon_a = \varepsilon_0 n_a^2$ that the cell is suspended as,

$$\vec{F}_T(r) = -8\left(n_a{}^2 - n_b{}^2\right)\varepsilon_0 \pi \frac{E_o^2 w^4}{r^2} e^{-\frac{(r^2+R^2)}{2w^2}} \left[ \frac{Rr}{w^2}\cosh\left(\frac{Rr}{w^2}\right) - \sinh\left(\frac{Rr}{w^2}\right) \right] \hat{r} \tag{22}$$

The refractive index for the cell ($n_a$ = *1.545*) [27] is higher than that of the medium ($n_b$ = *1.33*) [28], and the trapping force is attractive.

### 3. Results and Discussion

A sample of selected successive images of the cell describing the post-ionization trajectory of the cell is shown in Figure 3. The horizontal red line connects the trapping point of the successive images. After the cell is ejected from the trap, its trajectory follows the polarization direction of the trapping laser for a perfectly aligned trap, which is shown by the green line. When the ionized cell gets ejected from the trap, a larger acceleration is observed for the treated cells than for the untreated ones (see Figure 3). The untreated cells have a larger number of frames when ejected because the treated cells disappear from the camera within a small number of consecutive frames (see Figure 3). Figure 3a illustrates the trajectory of the untreated cell and shows a smaller angle than the trajectory of the treated cell (Figure 3b). This is clear indication that DMDD affects the cells as it ruptures cell membrane, and reduces cell mass.

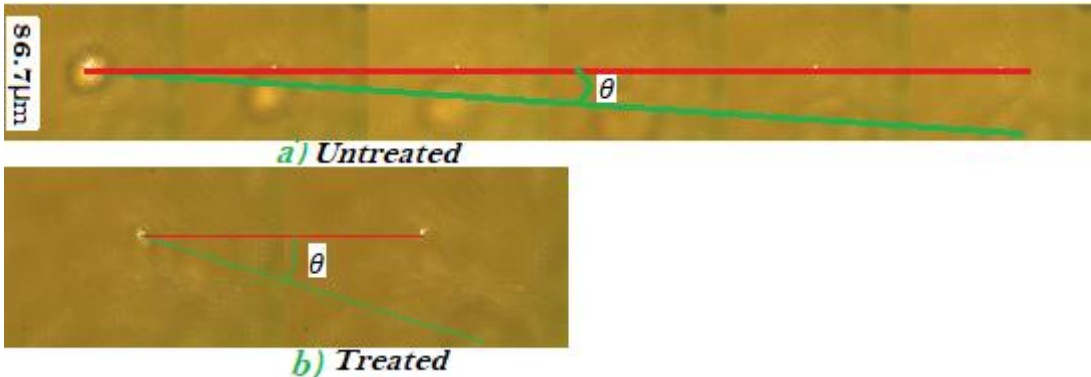

**Figure 3.** Motion of the (**a**) untreated and (**b**)treated ejected cells. The faster-ejected cells have a small number of consecutive frames with larger θ, and the slower-ejected ones have a larger number of consecutive frames with small θ.

The conventional understanding is that a cell gets ejected due to radiation pressure or misalignment. However, in this and previous studies [22,23], the cell remains trapped for up to 8 min, which eliminates the misalignment argument. The physically acceptable explanation is that a radiation field is causing cell ionization. The ionization process gradually leads to a buildup of charge in the cell resulting in a time-dependent charge density, $\rho(r,t)$. Thus, the oscillating electric field, $E_0 cos\omega t$, leads to an increasing electrical force, $F = \int \rho(r,t)E_0 cos\omega t \, dv$, which eventually overcomes the dominant intensity gradient trapping force and causes the cell to be ejected along the polarization direction of the laser beam [20–24].

This trajectory is defined by the trap, the drag, and the electrical forces acting on the cell according to Equation (1). For the post-ionization dynamics, we are interested in two electrical properties for the 4T1 cells in the untreated control, 2 h, and 24 h-treated groups. These properties include the net charges developed on the cell and the change in refractive indices due to the dielectric breakdown occurring when the cells are ionized. Using two different approaches, we studied these properties.

In the first approach, we determined each cell's charge and refractive indices; we considered a negligible change in the refractive indices. In the second approach, we determined only the charge based on the assumption that the refractive indices of the cells are the same for both treated and untreated cells.

### 3.1. Reduced Method

The statistical reduction method is a procedure that allows the extremes and outliers of data to be ignored, as they are far from the majority and thus unimportant. We consider the reduced data for the charge and the refracted index. In this statistical reduction method, the reduced data are obtained by eliminating data points far off from most data for the values of the charge-to-mass ratio and the radius. For example, for the radius value, three from the top and three from the bottom for the control group, five from the top and five from the bottom for the 2 h treatment, and nine from the top and nine from the bottom for the 24 h treatment were eliminated. These reduced data are sorted further by the radius in ascending order and sub-grouped. Similarly, the charge-to-mass ratio values eliminated six maxima and three minima for control groups, seven maxima and five minima for 2 h-treated, and three maxima and three minima from 24 h-treated groups. These reduced data are sorted further by the radius in ascending order and sub-grouped.

Additionally, we first sort each datum by the TIE or TRD in ascending order and eliminate the two minimum and the two maximum values from each of the three groups. We then conducted another sorting by mass in ascending order and again eliminated the two minimum and the two maximum values. The reduced data were obtained following

this procedure for the TIE and the TRD. Further reduction was made by sub-grouping the data with mass increments and calculating each subgroup's average mass, TIE, and TRD.

### 3.1.1. First Approach

The post-ionization displacement of the 4T1 cells for the three groups as a function of time is shown in Figure 3. The trajectory of a cell ejected from the trap follows the trapping laser's polarization (see Figure 2) and is determined by solving Equation (1). Lim et. al. (2004) reported that a cell becomes dislodged when the stage moves as a continuous unit to overcome the trapping forces [29]. In our case, this dislodgement or ejection of the cell (or multiple cells) occurs after about 5 to 8 min under the intense ionization beam. Figure 3 shows the path of the cells taken frame by frame as it moves perpendicularly to the beam direction. This means that the trapping force was overcome. Based on the average maximum displacement and the average size of the traveling cells, we can approximate the electrostatic force in Equation (11) and the trapping force in Equation (22). The average measured radius of the 4T1 cells is about $R = 8.2 \ \mu m$ [27] and the average measured maximum displacement is less than $r = 50 \ \mu m$.

Figure 4 is a displacement vs. time graph for the data of untreated, 2 h-treated, and 24 h-treated cells. The data are scattered in a broader range as a result of the mass variation of the 4T1 cells. The ionized cells have different sizes, from small to large, since we cannot select a similar size. This large variation in mass causes the data to become spread. On the other hand, the beam radius calculated at the trap location using this average size of the cells is $w$ = 282.26 μm. Using these values, we found $r^2/w^2 \simeq 0.03$, $Rr/w^2 \simeq 0.005$ and $(r^2 + R^2)/w^2 \simeq 0.03$. Applying the series expansions,

$$e^{-x} = 1 - x + \frac{1}{2}x^2 \ldots, x\cosh(x) - \sinh(x) = \frac{1}{3}x^3 + \frac{1}{30}x^5 \ldots \tag{23}$$

In Equations (11) and (22), and by keeping only the first-order terms, we can approximate the electrostatic force as

$$\vec{F}_e\left(\vec{r}\right) \approx q_0 E_0 \hat{s}_0 \tag{24}$$

and the trapping force as

$$\vec{F}_t\left(\vec{r}\right) \approx -kr\hat{s}_0 \tag{25}$$

where

$$k \approx 8\pi R^3\left(n_a{}^2 - n_b{}^2\right)\varepsilon_0(E_0/w)^2/3 \tag{26}$$

is a constant that depends on the electric field amplitude at the trap location (or the power), the beam and the cell radii, and the difference in the refractive indices between the cell and the medium. From the results in Equations (13), (24), and (25), the equation of motion for the cell given in Equation (1) can be written as

$$\frac{d^2r}{dt^2} + 2\gamma\frac{dr}{dt} + \omega^2 r(t) = \frac{q_{0E_0}}{m} \tag{27}$$

where

$$\gamma = 3\pi\mu R/m \tag{28}$$

and

$$\omega = \sqrt{8\pi E_0{}^2(n^2 - n_0{}^2)R^3\varepsilon_0/3mw^2} \tag{29}$$

We determine the mass of each cell using the measured value for the radius R and the density of the cell. There have been several approaches devised to measure cell density [30–32]. We considered a spherical model for cancer cells with $m = 4\rho\pi R^3/3$. Equation (29) becomes

$$\omega = \sqrt{2\varepsilon_0 E_0{}^2(n^2 - n_0{}^2)/\rho w^2} \tag{30}$$

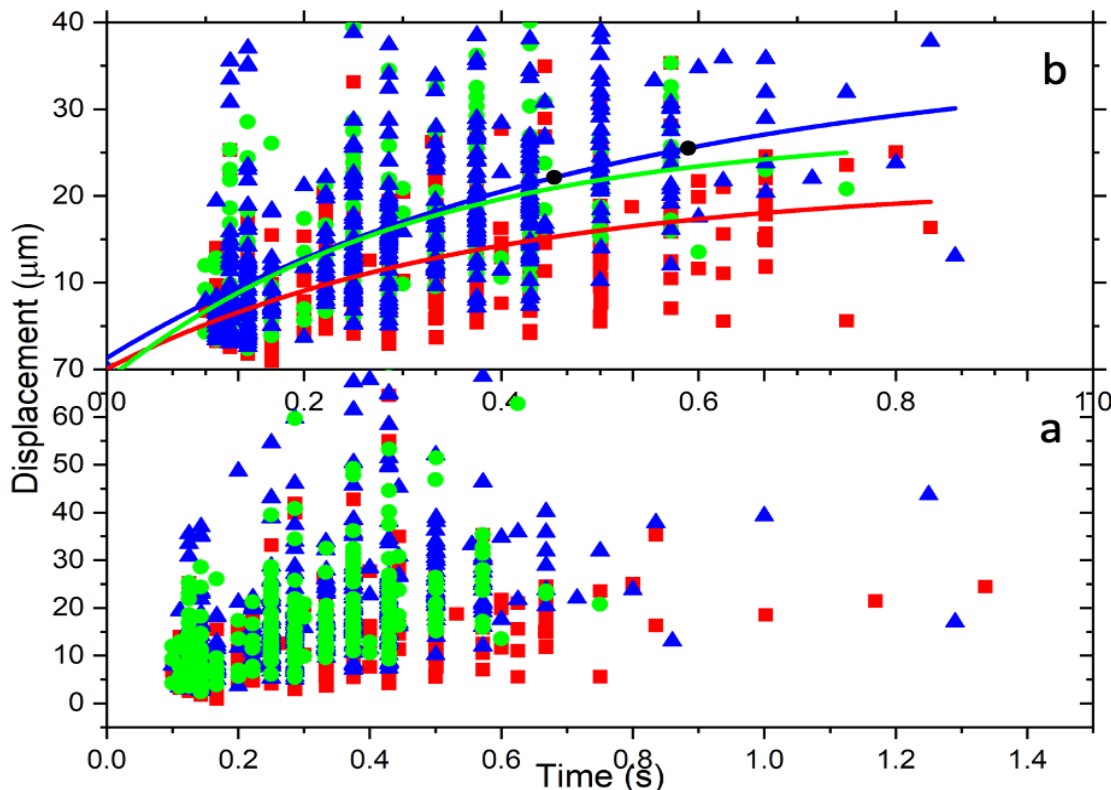

**Figure 4.** The radial displacement is a function of time. (**a**) All cells' displacement over time and spread according to the cell's mass. (**b**) These are reduced data resulting from the method described above. The data are fitted for a particular mass using Equation (31). The colors represent control (red), 2 h-treated (green), and 24 h-treated (blue).

Equation (27) describes an electrically driven damped harmonic oscillator. Under the approximation that the cell has no initial velocity and is positioned at the center of the trap, the solution to Equation (27) is found to be

$$r(t) = \left(qE_o/m\omega^2\right)\left\{1 - exp(-\gamma t)\left[\cosh\left(\sqrt{\gamma^2 - \omega^2}t\right) + \left(\gamma/\sqrt{\gamma^2 - \omega^2}\right)\sinh\left(\sqrt{\gamma^2 - \omega^2}t\right)\right]\right\} \tag{31}$$

In Figure 4, the post-ionization trajectory of the cells is characterized by an overdamped harmonic oscillator, which requires $\gamma^2 \geq \omega^2$. As a result of the ionization caused by the radiation, the cell underwent structural changes that changed its electrical susceptibility. This led to a new refractive index, $n < 1.545$, that must be greater than the refractive index of the surrounding medium (FBS, $n_0 \approx 1.33$). To find this new refractive index of the total free charge, the maximum value $\omega_{max} = \gamma$ was determined using the values for $m$ and $R$ for each cell and $\mu$ for FBS [33]. The corresponding maximum refractive index, $n_{max}$, was determined using the relation for $\omega$, the electric field amplitude, $E_o$ = 42.72 kV/m, determined from the measured power and the beam radius, $w$ = 282.26 μm, at the trap location. The numerical model fitting function, NonlinearModelFit in Mathematica, was used to fit Equation (31) to the displacement versus time data shown in Figure 4 for each cell. The NonlinearModelFit function started looking for the charge $q$ and the refractive index $n$ at several orders of magnitude below $n_{max}$. The charge is expressed in the units of electron number by dividing the charge of a cell by the electron charge. The results for $q$ measured by the $z$ number ($z = q/(1.6 \times 10^{-19}$ C)) and $n$ are shown in Figure 5.

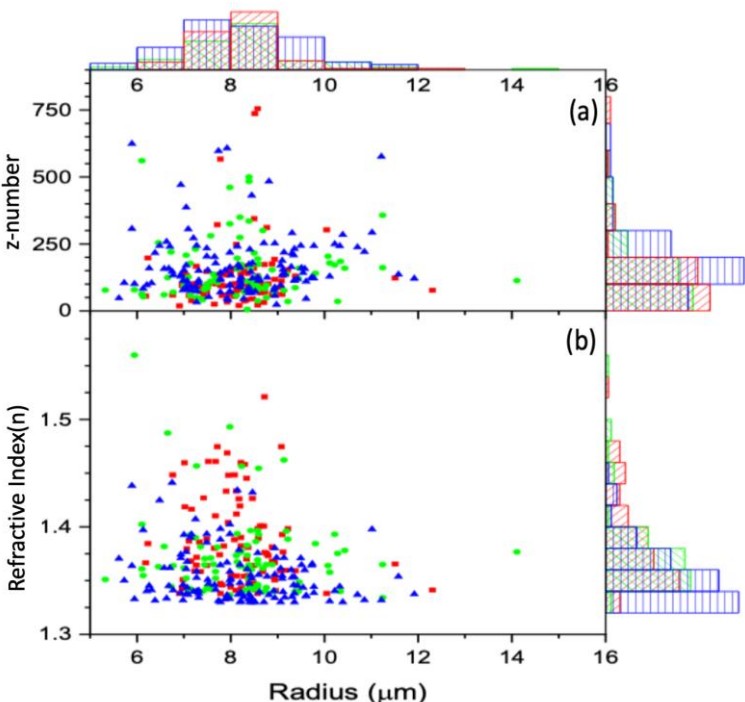

**Figure 5.** The calculated z number (**a**) and refractive index (**b**) vs. the radius for each cell, along with the corresponding distributions, are displayed using histograms: control (red), 2 h-treated (green), and 24 h-treated (blue). Both the refractive index and z number are unitless.

Figure 5 shows the z number and the refractive indices variation as a function of cell size. For all the cells, the z number versus radius for the untreated, 2 h-treated, and 24 h-treated is shown in Figure 5a. Figure 6 displays the reduced data obtained from Figure 5a, first sorted by radius in ascending order and eliminating three from the top and bottom, then sorted by charge in increasing order and removing three maxim and three minima. These reduced data are sub-grouped by 0.16 µm increments. The average for each subgroup was calculated, and the resulting data for z vs. R are shown in Figure 6. According to the result in Figure 6, the bigger the cell is, the higher the charge.

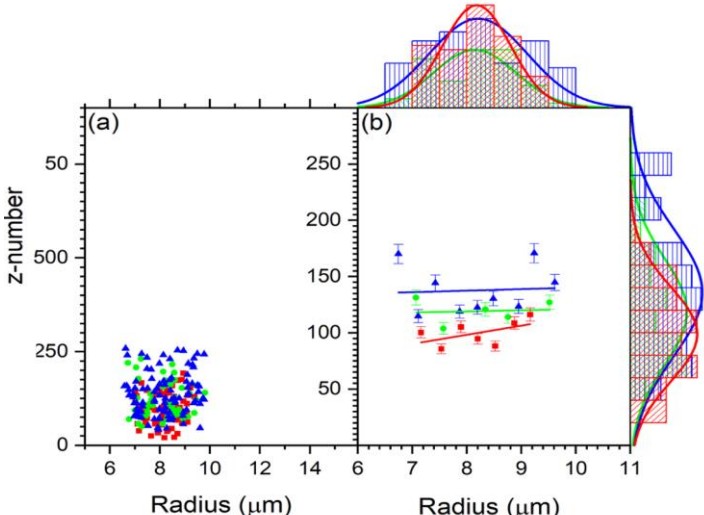

**Figure 6.** (**a**) The scatter plot of the charge distribution with size for control (red), 2 h-treated (green), and 24 h-treated (blue) group. (**b**) Reduced data for the charge vs. R using Origin data manipulation reduction by grouping for control (red), 2 h-treated (green), and 24 h-treated (blue). The charge is measured by the z number.

For all the cells, the refractive index versus radius for the untreated, 2 h-treated, and 24 h-treated cells is shown in Figure 5b. Figure 7 displays the reduced data obtained from Figure 5b in the same way as Figure 6. The average for each subgroup was calculated. The resulting data for n vs. R are shown in Figure 7. According to the result in Figure 7, the refractive index stays unchanged with the treated cells and decreases with a radius for the untreated cells.

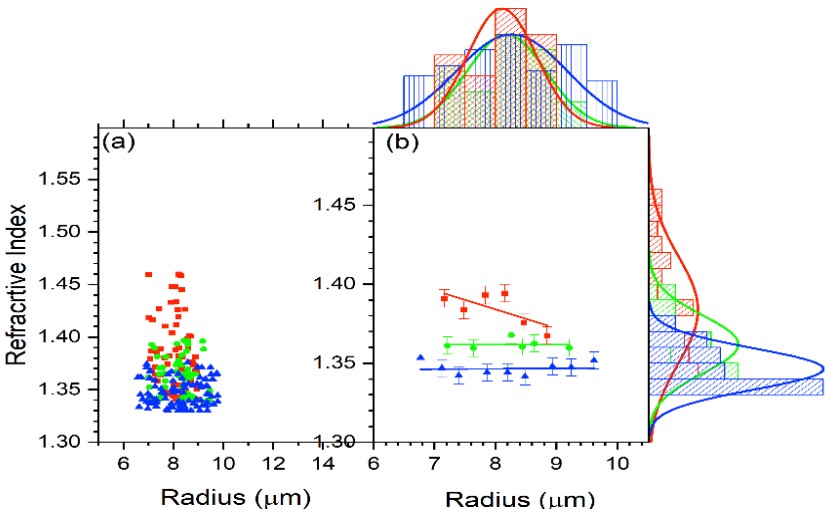

**Figure 7.** (**a**) The refractive index behavior in relation to cell radius for the control (red), 2 h-treated (green), and 24 h-treated (blue) group. (**b**) Reduced data for the refractive index vs. R using Oigin data reduction method by grouping and then fitted linearly for control (red), 2 h-treated (green), and 24 h-treated (blue) group.

The z numbers were found to be z = *117+/−56, 129+/−61,* and *156+/−68,* and the refractive indices were n = *1.382+/−0.041, 1.377+/−0.040,* and *1.357+/−0.026* for the control, 2 h-, and 24 h-treated groups, respectively (see Figure 8). For all three groups, the results for the charge show a higher standard deviation, while the refractive indices show a negligibly small standard deviation. This suggests it is worthwhile to examine the variation in charge caused by cell size and determine if there is a relationship between refractive index and charge.

### 3.1.2. Second Approach

If we neglect the change in the electrical susceptibility of the cells after ionization, we may use the accepted value for the refractive indices of untreated cancer cells, $n = 1.545$, and for the medium, $n_0 = 1.33$. Using these values for the trapping force in Equation (22), the electrostatic force in Equation (2), and the drag force in Equation (12), the equation of motion in Equation (1) can then be written as

$$m\frac{d^2r}{dt} = q_0 E_0 exp\left(-\frac{r^2}{w^2}\right) - 6\pi\mu R\frac{dr}{dt}$$
$$-1.236\epsilon_0\pi\frac{E_0^2 w}{r^2}e^{-\frac{(r^2+R^2)}{2w^2}}\left[\frac{Rr}{w^2}\cosh\left(\frac{Rr}{w^2}\right) - sinh\left(\frac{Rr}{w^2}\right)\right] \quad (32)$$

The charge is determined by the NonlinearModelFit.

Figure 9b displays the data reduction first sorted by radius in ascending order, eliminating three from the top and bottom, then sorted by charge in increasing order, and deleting three from the minimum and maximum. Finally, Figure 9c shows the reduced data displayed in 9b sorted by radius in ascending order and reduced by grouping with 0.20 x-increments by Origin Pro. As clearly shown in Figure 9c, the charge increases as the radius of the cell increases. Figure 9d displays the charge as a function of time for each untreated, 2 h-treated, and 24 h-treated cell. In Figure 9f, we sorted by charge and removed the charge

of three cells from the top and bottom, and then the data reduction by grouping was made by sorting in time, which clearly shows that the charge on the cell decreased after ejection as the time increased. The charge decreases with time in a very similar way for all three groups and is smaller for the untreated samples than for the treated ones. The decreasing charge on the cell is due to the interaction of cells with the environment outside the trap.

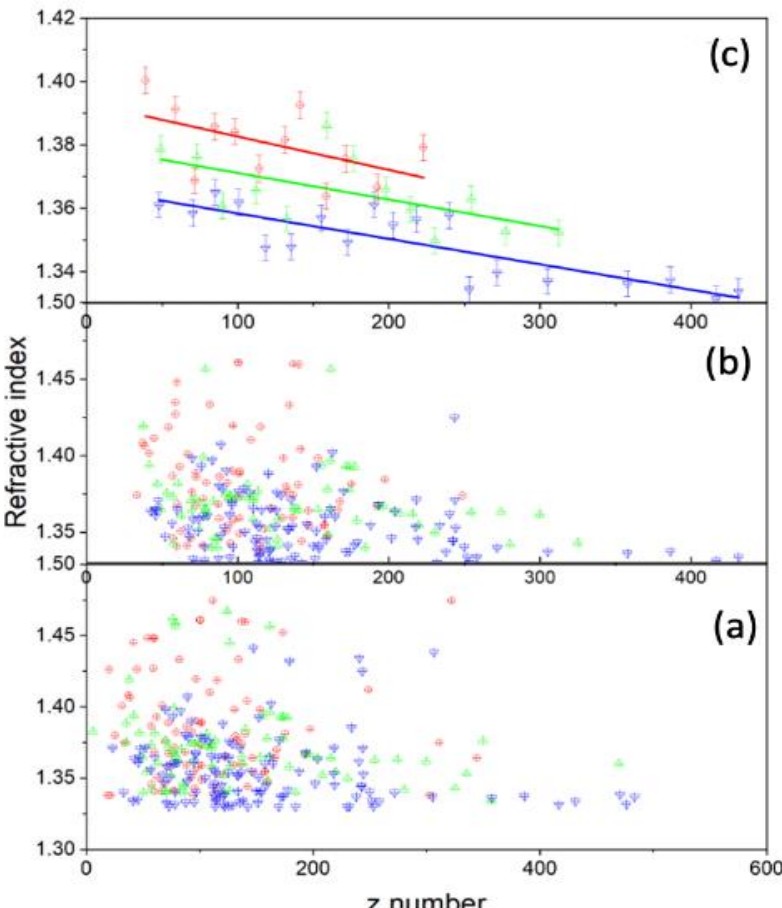

**Figure 8.** (**a**) The scatter plot shows how the refracted index relates to the charge. (**b**) This is after the reduced method was performed on the data. (**c**) The data after further reduction was carried out and then fitted. There is a negative correlation between charge and refracted index.

Additionally, the charge calculated was smaller than in the first model. This is because of the trapping force for the treated cells being calculated with the electric permittivity ($\epsilon$) of the untreated cancer cells. However, the electric permittivity of the cells after a dielectric breakdown differs from the unirradiated samples since dielectric breakdown increases the conductivity of the cell, and eventually, the permittivity decreases.

### 3.2. Radiation Effect on the Cell Charge and Refracted Index

The statistical distributions for charge and refractive index vs. the threshold radiation dose (TRD) and the threshold radiation energy (TIE) for all cells in each group are displayed using histograms in Figure 9a–d. To compute TIE and TRD, we determined the average power incident on the cell, which remained the same during the ionization of each cell, to be $P_I$ = 0.8064 W, and the estimated transmitted power was $P_T$ = 0.64 W.

$$TIE = \frac{A_{cell}}{A_{beam}}(P_I - P_T)T \tag{33}$$

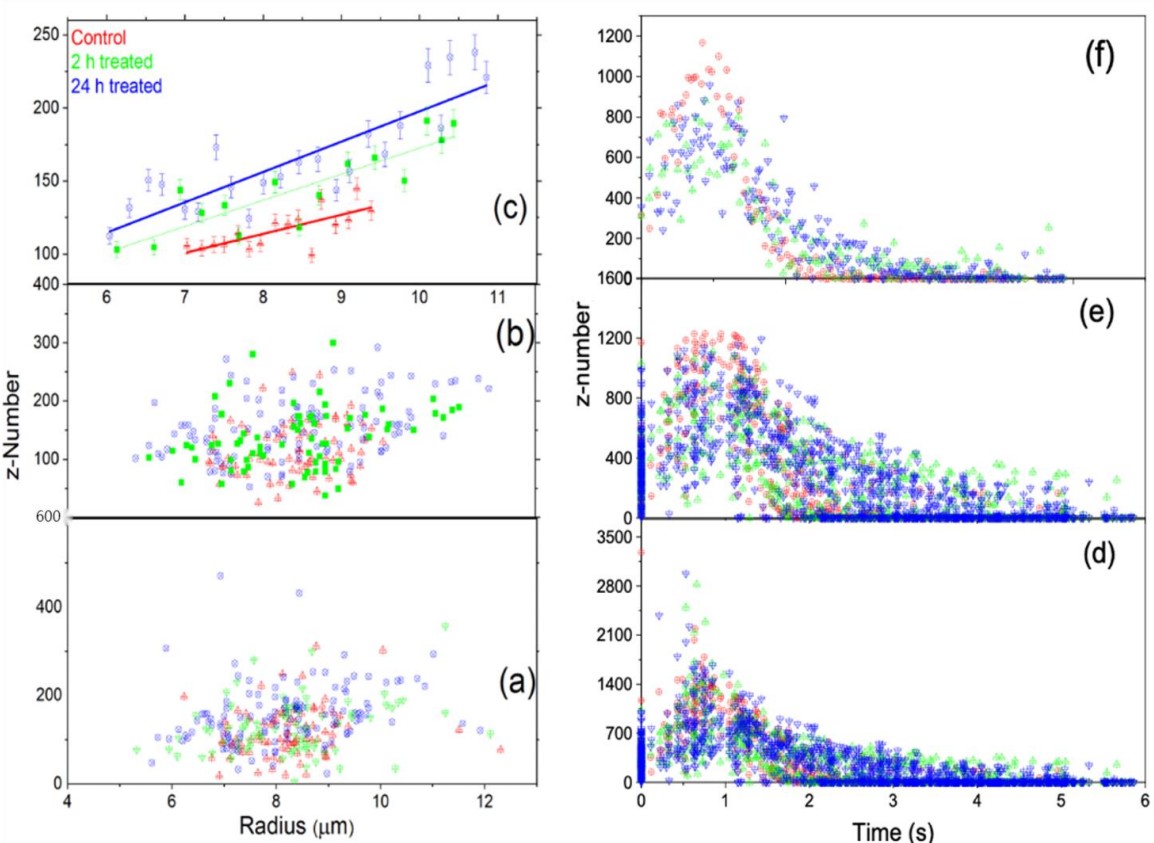

**Figure 9.** The data distribution of the whole untreated, 2 h-treated, and 24 h-treated data for the three groups. (**a**,**b**) show the behavior of the charge with radius. (**c**) The data is reduced further from (**b**) using Origin data manipulation reduction by grouping method, and linearly fitting. (**d**–**f**) show that charge decreases with time.

In Equation (32), *T* is the ionization period determined using the digital camera image grabbing rate and the number of images captured during the time covering the instant each cell entered and the instant it was ejected from the trap. *Abeam* is the beam size determined at the trap location using the numerical aperture of the objective lens [20]. The TRD was then calculated for each of the 4T1 cells using

$$TRD = \frac{TIE}{M_{cell}} \tag{34}$$

In each graph, the data coded red represents the untreated control group, green denotes the 2 h-treated group, and blue denotes the 24 h-treated 4T1 cells. From these distribution graphs and the calculated average values, the TRD and TIE vs. charge and refracted indices for the treated groups are less than the untreated control group. The effect is amplified by increasing treatment duration, as evident from the lower TIE for the 24 h-group when compared to the 2 h-group (see Figure 10).

Low TIE is required for ionizing cells with small refractive indices to build a sufficient charge for the cell to escape the trap. For a cell with a high refractive index, there is a greater need to ionize it so that it can break down its membrane and develop a large enough charge to push it out. Therefore, as the refractive index increases, the TIE needed to ionize the cell must also increase. If a cell has a fixed TIE of magnitude "*x*", and $n_1 > n_2$, then a cell with $n_2$ will develop a greater magnitude of charge than a cell with $n_1$. Due to their low refractive index, loosely attached molecules require a lower TIE and will develop larger charges when ionized. In a similar sense, molecules that are tightly bonded have a larger

refractive index and require a larger TIE, so when the molecules are ionized, a smaller amount of charges are generated.

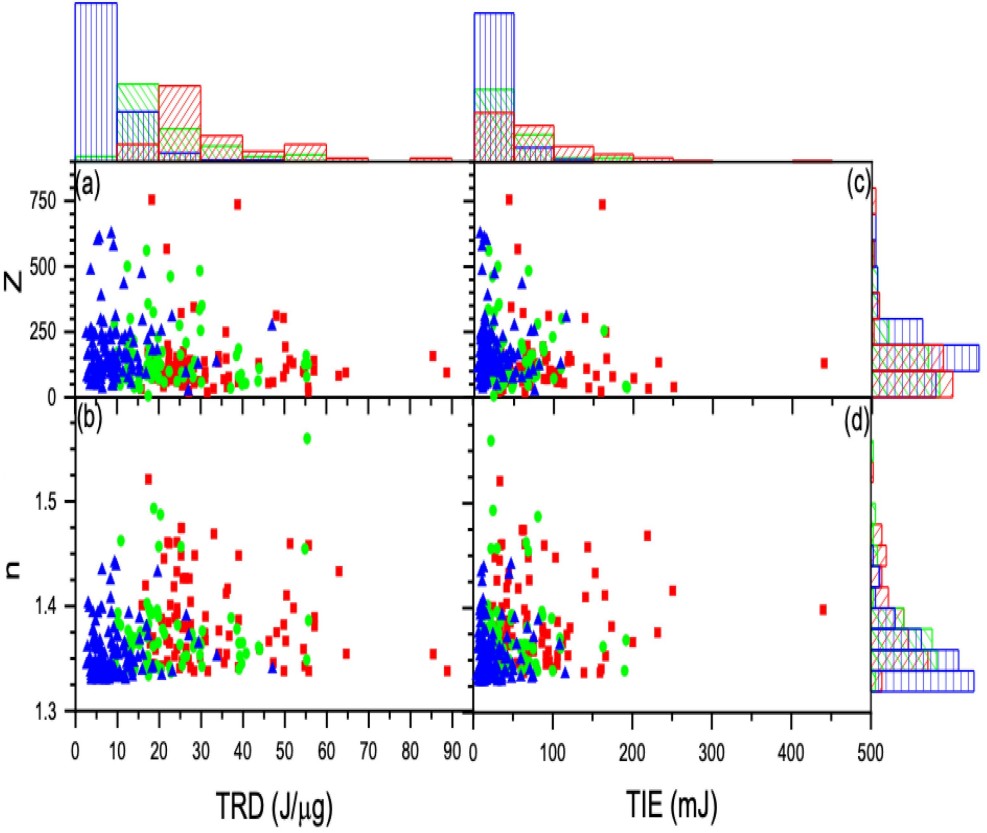

**Figure 10.** The calculated charge and refractive index vs. TRD (**a**) and (**b**) vs. TIE (**c**) and (**d**) along with the corresponding distributions displayed using histograms: control (red), 2 h-treated (green), and 24 h-treated (blue). The charge is measured by the z number.

Moreover, the relationship of charge with the trapping constant $k$ depends on the permittivity of the cell. The reduced data method described above was applied, and the result is displayed in Figure 11. As shown in the top graph of Figure 11, the spring constant increases with the charge on the cell; according to Figure 11, untreated cells also have a greater slope value than 2 h-treated or 24 h-treated cells. Thus, we can draw from this behavior that treatment causes the cell to become more elastic, such that the cell can accommodate charge build-up, which also explains the reason why the treated cell exits the trapped at a higher angle, as illustrated in Figure 4.

The radiation incident on the cancer cells delivers the right amount of ionization energy to result in dielectric breakdown. When the breakdown of the cell membrane increases, the conductivity and charge density increase. Thus, a material's conductivity is closely related to the ability of its charge to be transported through its volume by an applied electric field. The permittivity describes the material dipoles' ability to rotate or for its charges to be stored in response to an external field [34]. An external field causes extreme polarization, leading to torque due to the misalignment of induced dipoles. The membrane structure is rearranged to form aqueous pores that increase the membrane's conductivity and permeability so that water molecules can pass through the membrane into the cell (reversible electroporation) [35–37]. The strong rapid oscillating electric field causes the membrane to not reseal, such that electroporation becomes irreversible. Electrons are permanently dissociated from atoms, causing the cell to ionize.

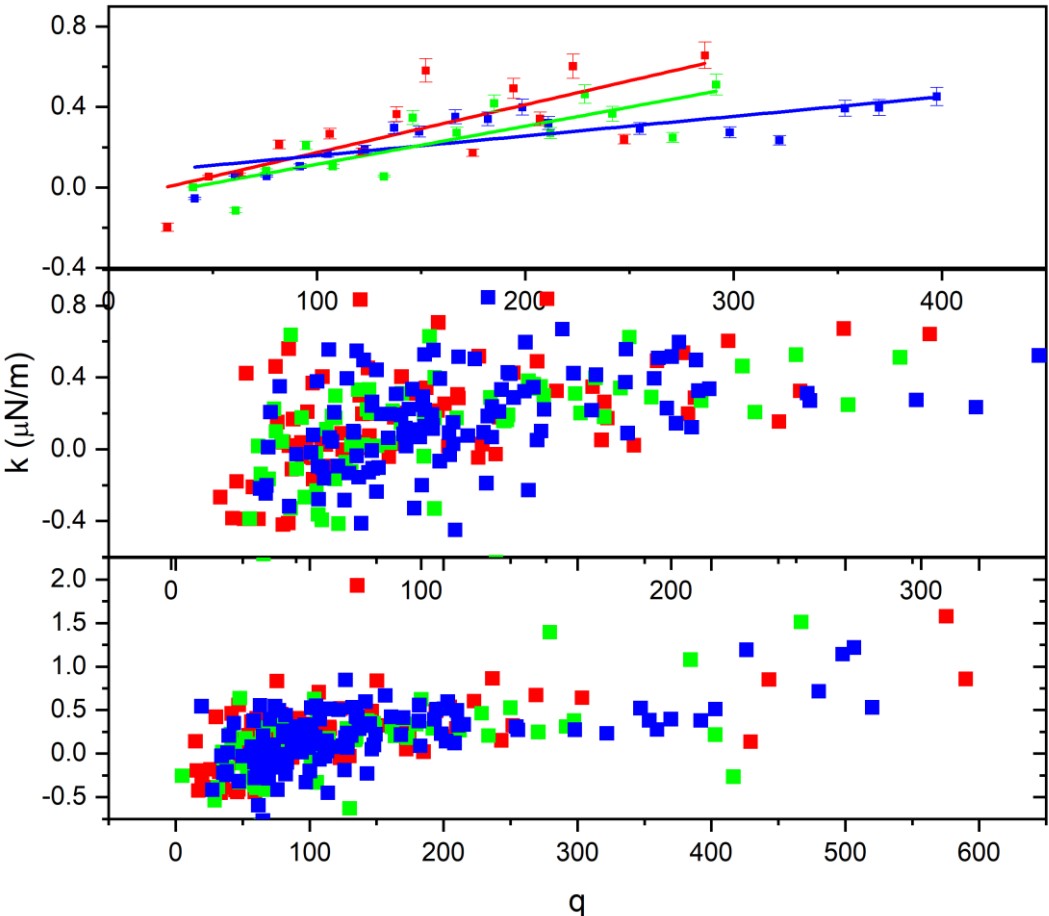

**Figure 11.** The relationship between the spring constant and the charge developed on the cell.

Furthermore, the drug initially damages the membrane of the 4T1 cells, which is further damaged as it is exposed to the laser's field. This oscillating field causes the temperature of the cell to rise, which means liquids become less dense and less viscous. Stogryn showed that when the temperature of the sodium chloride solution increased, the dielectric constant decreased, supporting our result [38]. Our result is also in conformity with the observation that when the conductivity of a material increases, its permittivity decreases [39].

Additionally, the charge building up at low TIE for the treated cells means less exposure time. Since the TRD is the incident ionization energy per cell mass, this implies that the TRD will be smaller for the treated cells [20,40]. The untreated cells require a high radiation dose, which means that cells are exposed longer at higher TIE, causing greater cell damage to the healthier cells.

Similar behavior was observed with the refracted index vs. TIE. The refracted index was higher at low ionization for the treated cells and fell rapidly with increased TIE. However, in the case of the untreated cells, it was constant with increased radiation (see Figure 10c). The cells that were treated for 24 h had a much smaller spread with increased ionization energy. The longer the cells were treated, the narrower the spread of both charge and refracted index. We know that the refractive index of any material depends on its chemical composition, which dictates its electrical and magnetic properties [41,42]. We also found that as the 2-Dodecyl-6-methoxycyclohexa-2, 5-diene-1, 4-dione (DMDD) diffused into the cells, it appeared to alter their chemical composition so that their refractive index decreased, weakening the cells. Additionally, it was previously demonstrated that since these DMDD molecules had a typically long shape, when placed in the laser field, electrons would more easily oscillate parallel to them than perpendicularly [43]. As the incident

radiation's electric field polarizes the molecule it falls upon, an oscillating dipole is formed. The oscillating electric field of the incident radiation creates an oscillating electric dipole such that the electric field around it oscillates. This results in an oscillating electric field, influencing the oscillating radiation field [44]. This new electric field develops because of the addition of charges, which interact with the charges in the material, changing their optical properties. As such, we conclude that oscillating ionizing radiation causes apoptosis at low energy.

## 4. Conclusions

This study investigated the behavior of cell charge and refractive index after treatment with DMDD, a naturally occurring antitumor compound derived from the roots of Averrhoa carambola to treat 4T1 breast carcinoma cells. We have calculated the post-ionization charge (measured by the z number, which is the charge/charge of an electron) and the refractive index for the control untreated, 2 h-, and 24 h-treated 4T1 breast cancer cells. The results show that the charge increases while the refractive index decreases with the length of time cells are treated. Both the charge and the refractive indices for the control and treated groups seem to be essentially the same.

Furthermore, the radiosensitive nature of 4T1 cells was determined by comparing the TIE and TRD measured in vitro after 2 h- and 24 h-treated with an untreated control group. We demonstrated a new technique for determining TIE and TRD that uses laser trapping for ionizing single and multiple cells. Notable differences were observed between treated and untreated cancer cells. The results obtained clearly demonstrate an increase in the radiosensitivity of the 4T1 cells due to the antitumor compound DMDD [20]. As a result of radiation-mediated electrical and thermal interactions within the cells, the importance of induced charges and hyperthermia is demonstrated. In addition to the effect stemming from the antitumor compound used to treat the cells, the significant reduction in the TRD in multiple-cell ionization is associated with the chain effect of ionization by the radiation field and the absorption by water molecules at 1064 nm [40].

It is important to point out that, generally, the results reported in this study highlighted the effect of combined modalities in radiotherapy, chemo, and hyperthermia useful only for in vitro cancer treatment. This model provided an effective way to compute the change in the cell's refractive index and charge because of the oscillating field. One important prospective study is an accurate measurement of the charge and the temperature elevation that occurs when the cells interact with radiation after a treatment.

**Author Contributions:** Conceptualization, D.B.E. and E.M.; methodology, D.B.E., H.T.C., L.C. and E.M.; software, H.T.C., D.B.E., C.N. and E.M.; validation, D.B.E., H.T.C. and E.M.; formal analysis, D.B.E., H.T.C., C.N. and E.M.; investigation, E.M., M.K., C.V., M.G., Y.G. and L.C.; resources, D.B.E., Y.G. and L.C.; data curation, E.M. and H.T.C.; writing—original draft preparation, H.T.C., D.B.E. and E.M.; writing—review and editing, H.T.C., E.M., C.N. and D.B.E.; visualization, H.T.C., D.B.E. and E.M.; supervision, H.T.C. and D.B.E.; project administration, H.T.C. and D.B.E.; funding acquisition, H.T.C. All authors have read and agreed to the published version of the manuscript.

**Funding:** This research received no external funding.

**Institutional Review Board Statement:** Not applicable.

**Informed Consent Statement:** This article does not contain any studies with human participants performed by any of the authors.

**Data Availability Statement:** The datasets generated during and/or analyzed during the current study are available from the corresponding author upon reasonable request.

**Acknowledgments:** The authors also thank Renbin Huang at Guangxi Medical University for providing the DMDD sample.

**Conflicts of Interest:** The authors declare that there are no conflicts of interest.

**Statement of Significance:** The study of the refractive index of a cancer cell is the study of its dielectric constant. The dielectric constant of a cancer cell determines its ability to store electric charge, which tells us how the cell holds the amount of electrical energy before its breakdown. Thus, the post-ionization dynamic quantities such as displacement, velocity, and acceleration depend on the charge. Further, the dielectric strength relates to the refractive index of the cancer cell, which determines the maximum electric field that the cell can withstand. To stop tissue toxicity caused by radiation and enhance cancerous cells' sterilization process, a favorable tradeoff between treatment benefit and morbidity must be made. To destroy cancerous cells effectively, radiation must also be balanced by preventing the side effects of radiotoxicity on healthy cells. In general, the dielectric constant (refractive index) determines the maximum electric field required for cancer cell apoptosis.

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
