# Peer review of "Measurement of Charge and Refractive Indices in Optically Trapped and Ionized Living Cells"

_tomography, doi:10.3390/tomography9010007_

Round 1
Reviewer 1 Report
The paper is well presented and the results are interesting. I recommend publication after correction of a few minor typographical errors and correction of some formatting issues.
Here are a few things I caught:
Fig. 1 caption seems to be missing a "W"
Line 316+ is a little unclear, perhaps restate it
Line 331 replace was with were
List the units used for charge in caption or axis label in Fig (5), etc
Figs 6-10 should be larger to make them easier to read
Line 474 unit should be units
Line 486 sort should be sorted
Author Response
We would like to take this opportunity to thank the reviewer for their diligence in reviewing our work.
Point 1: Fig. 1 caption seems to be missing a "W".
Response 1: This was fixed; see line 429 in the word file.
Point 2: Line 316+ is a little unclear, perhaps restate it.
Response 2: This was rewritten; see line 637 in the word file
Point 3: Line 331 replace was with were.
Response 3: This was fixed; see line 675 in the word file
Point 4: List the units used for charge in the caption or axis label in Fig (5), etc.
Response 4: This was addressed. The charge was replaced with z-number, which is a unit less quantity. See line 862 in the word file. Further in the caption, we stated that the z-number was unit less. See line 862 866
Point 5: Figs 6-10 should be larger to make them easier to read.
Response 5: Images were made larger. See lines 889, 897, 905, 933, and 984
Point 6: Line 474 unit should be units.
Response 6: This was fixed; see line 837 in the word file
Point 7: Line 486 sort should be sorted.
Response 7: This was fixed; see line 940 in the word file.
Changes were made to accommodate better readability, as requested by reviewer 2. This is shown by track changes in the the word file.
Reviewer 2 Report
The manuscript should be modified so that the language is simpler for a non physics expert to understand. Relevance to the broader scientific community and particularly the oncology community should be specified throughout the M/S.
Author Response
We would like to thank the reviewers for their spending in reviewing this manuscript.
Point 1: The manuscript should be modified so that the language is simpler for a non physics expert to understand. Relevance to the broader scientific community and particularly the oncology community should be specified throughout the M/S.
Response 1: The reviewer's suggestion of making the manuscript more readable for non-physics was taken seriously.
The track changes shown in the manuscript show the effort we expend in following the reviewer's advice. We understand that the subject presented here is mathematically technical for non-physics readers. Nevertheless, we believe the findings are clear for non-physics readers. With the reviewer's recommendation, we went back through the manuscript with the intention of better readability.
As a result of this work, the research community will better understand the role biophysical parameters play in developing better cancer treatments.
Changes were made to accommodate better readability, as requested by reviewer 2. Please see the Track changes in the word file.